

# Research on the perception of the terrain image of the tourism destination based on multimodal user-generated content data

Tao Hu and Juan Geng

School of Tourism, Hainan University, Haikou, China

## ABSTRACT

Destination image is a powerful means by which destinations compete in the tourism industry, and the accurate identification of a destination image better serves destination marketing and management. This study uses multimodal data, such as text, images, and videos uploaded by tourists, to construct a comprehensive and systematic destination image process. The "cognitive-emotional-overall image" model, latent Dirichlet allocation (LDA) model, and deep residual neural networks are implemented to build a framework to examine the perception of a destination image, travelogues, and short videos from the sources called Ctrip, Qunar, and TikTok. The results show that tourists' overall perception of Sanya is based mainly on the cognitive image of natural scenery, human resources, and food. In addition, there are differences between textual and visual cognitive images among the perceptual images when multimodal data is under consideration. Furthermore, tourists have an overall positive affective image of Sanya as a destination.

## INTRODUCTION

The image of a tourism destination plays a vital role in marketing, brand differentiation, and preference of a destination in business (*Zhou, Wei & Liang, 2013*; *Gao & Jiao, 2014*; *Zhao, Zhu & Hao, 2018*). Most studies working on the decision process of choosing a destination have pointed out that destination image, perceived quality, and the price of tourism products or services are the main drivers of destination choice. Moreover, tourists' perceived tourism image is essential for a destination since it influences potential tourists' travel behaviors, satisfaction, and decision-making (*Zhang et al., 2019*). Therefore, researching and understanding tourism destination image can help destinations to gain core competencies.

The development of Information and communications technology (ICT) has shifted marketing strategies involving mass marketing evolution to those involving data-driven marketing. At the same time, it has enabled tourists to share their travel experiences through text, images, or videos anytime and anywhere. User-generated content (UGC)

Corresponding author
Juan Geng, gengj@hainanu.edu.cn

**OPEN ACCESS**

shapes visitors' perceptions of destinations. Along with the rapid development of artificial intelligence and the accelerated implementation of 5G commercialization, short video software, for example, TikTok, is becoming increasingly popular among travelers. The ability to generate many short, user-generated videos with text content on the web through multimedia has led to the exponential growth of data resources. User-generated data have gradually evolved from unimodal data with text-based information to multimodal data combining images, video, and text. Moreover, multimodal analysis has become a powerful tool for multimedia content analysis and understanding. Multimodal data can help us describe objects or phenomena from different aspects or perspectives using complementary or supplementary data information. Showing that using multimodal data can improve the performance of the analysis (*Rahate et al., 2022*).

In the available studies on tourism destination images, web-based text analysis has been mainly utilized (*Wanf et al., 2013*; *Chen, Zhang & Du, 2014*; *Lu & Liao, 2019*; *Tan, Liu & Li, 2021*). User-uploaded images and video information, which is rich extractable information, have been ignored in data analysis. However, the big data revolution has brought about a new marketing dynamic by highlighting the interplay between advanced technologies (*e.g.*, machine learning and visual analytics) and consumer experiences on social media (*Dekimpe, 2020*; *Yu & Egger, 2021*). The implementation of computerized processes in this area has provided new approaches to tourism research, improving conventional techniques for analyzing UGC.

This article uses deep learning and natural language processing methods to study the integrated image perception of tourism destinations based on the ''cognitive-emotional-overall image'' theory, using multimodal data generated by tourists through multiple channels.

The rest of the article is structured as follows: Related work is presented in 'Related Work'. 'Design of the Research Framework' presented the proposed method. The results and discussion are allocated to 'Results and Discussion'. The research is concluded in 'Conclusion'.

## RELATED WORK

### Destination image

*Hunt (1975)* was the first to focus on destination image, arguing that it represents people's impression of a place in which they do not live. A model used for the formation of destination images, aiming to provide a research framework in which the construction of destination images is guided, was developed, suggesting the relationships between the different levels of the evaluation (cognitive, affective, and holistic) within its structure and the elements (*Baloglu & Mccleary, 1999*). In this context, cognitive and affective images are interrelated, with the affective image being largely dependent on cognition. Together, these two types of images constitute the overall image of a destination, influencing tourists' intention to visit and recommend it.

Early studies on the perception of a tourism destination image were based mainly on structured surveys of respondents, with questionnaires being the primary method to

obtain data. However, with the rapid development of 5G networks and self-media, many researchers have explored tourism destination images based on UGC data. Many scholars have researched destination images based on travelogue texts. *Xu, La & Ye (2018)* chose the review data of Nanjing tourists on Ma Hive and used extensive text data in the analysis. The study results showed differences between different dimensions of a tourism destination image, underlying which tourism attraction is the most critical. Moreover, *Lyu & Chen (2020)* used social network and content analysis to study the differences between tourists' perceived and promoted images in Henan Province. By taking traditional villages as an example, *Yuan et al. (2020)* selected user-generated text and image content to study the main attractors of traditional villages, which contribute to local tourism development. *Tan, Liu & Li (2021)* utilized content analysis and rooting theory-based methods to select travelogue reviews of Wutai Mountain on Ctrip to examine three types of images: tourists' cognitive, emotional, and overall images after the tour. In addition, several studies on tourism visuals in the analysis of destination images have gradually emerged. *Wang & Sparks (2016)* explored the gaze interests and visual trajectories of tourists by using eye-tracking techniques to identify the tourism images to which participants paid the most attention and the focus of these images. *Deng et al. (2019)* selected a Flickr dataset and employed a machine learning-based naive Bayes classifier to help DMOs select photo content, assign target photos, and address the apparent gap between the projected and received images. Furthermore, *Adel & Hamed (2019)* studied the extent of human presence in photos shared by tourists on Facebook and further analyzed the host-guest interaction in photos, finding that tourists were more inclined to take and share their photos and their travel companions. More up-to-date research can be found (*Lin et al., 2023*; *Zhou et al., 2023*; *Nan et al., 2023*)

## UGC and machine learning

UGC refers to content produced by travelers that shows assessments and opinions about products and services on internet platforms such as social media (*Zhao, Fang & Zhu, 2012*; *Yoo & Gretzel, 2011*), including online text data and online image data, among other data types. Due to the richness, diversity of sources, and authenticity of the data (*Yu & Sun, 2019*), user-generated online review data on social media can reveal tourists' preferences and sentiments as well as the popularity assessments of destinations and attractions (*Gandomi & Haider, 2015*; *Li, Huang & Christianson, 2016*) and has now become a source of scholarly and industry research and its credibility and vitalness increase (*Bigne et al., 2021*; *Kar, Kumar & Ilavarasan, 2021*; *Liang et al., 2021*). UGC is a crucial component of a destination's online image since it can reveal, among other things, tourist satisfaction with the destination. UGC directly impacts the perceived quality of potential tourists and their travel decisions. *Somabhai, Varma & Somabhai (2015)* argued that users can intuitively understand a destination by reading online travel reviews. The online image of a destination positively influences consumer behavior, increasing people's willingness to travel, revisit, or recommend the destination (*Huertas & Marine-Roig, 2016*). Moreover, *Marine-Roig & Clave (2016)* argued that UGC from social media is more authentic and trustworthy than that from other external sources (*Fotis, Buhalis & Rossides,*

*2012*; *Leung et al., 2013*), as it allows people not only to access information but also to understand real emotions towards the destination (*Chen et al., 2019*).

In tourism, studies have employed big data to reveal the novelty of UGC. For example, *Yu & Egger (2021)*; *Kuhzady & Ghasemi (2019)* indicated how picture features influence visitors' interactions with Instagram posts. However, only classification practices for travel photos were enhanced in that study, and data analysis remained limited to manual methods (*Yu & Egger, 2021*). However, advanced techniques such as machine learning and natural language processing (NLP) have enabled researchers to better understand visitors' behavior (*Vu et al., 2015*).

### Multimodality

Big data is multisource and heterogeneous. The study of single-mode data has limitations, but multimodal data analysis has become a new research hotspot. Moreover, the study of multimodal data can complement the rules of single-data information and expand the diversification of information research. In recent years, some studies have started using modal data other than text data to analyze tourism destination image perception. Computer vision algorithms employing deep learning methods allow for the analysis of visual-level features; *e.g.*, *Zhang, Chen & Li (2019)* implemented convolutional neural networks to examine photographs of tourists from different countries and to compare their perceptual differences. Visual content (*e.g.*, photos and videos) easily attracts users' attention due to its vividness and visual appeal, carrying a wealth of information. Some scholars have further selected data combining multiple modalities to carry out research. *Li, Huang & Christianson (2016)* chose photos with textual details and found that the textual information naturally embedded in tourist photos tended to attract more visual attention than other image components. *Sheng et al. (2020)* selected reviews about Xi'an on Ctrip and used text and image data to construct an image of Xi'an as a tourist destination.

Furthermore, *Li et al. (2022)* studied the emotional differences between two types of restaurant online reviews, photos, and text photos. They found that reviews with pictures were more valuable and exciting than those without photos. The former studies were more practical and exciting than the latter studies.

## DESIGN OF THE RESEARCH FRAMEWORK

This study focuses on using online texts and visual perspectives in the tangible medium of the tourist gaze. The realization of the tourist gaze is expressed through graphic texts. With the increasing development of the internet, tourists can easily and quickly find the travel information they need on various travel platforms and share their travel experiences. Therefore, this article analyses tourists' gazes with the help of crawlers and Gooseeker software to form Sanya's following tourism image constructs. The main research framework of this article is shown in Fig. 1.

### Research subjects and data sources

Sanya is a prefecture-level city in Hainan Province located at the southernmost tip of China's Hainan Island and is an international tourist city with tropical seaside scenery (*Baidu, 2023*).

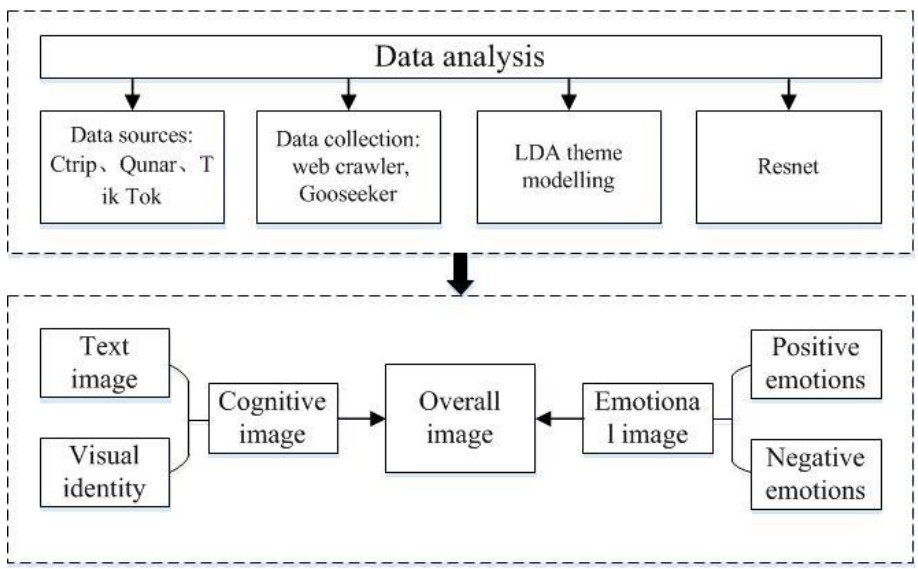

**Figure 1   Main flow chart.**

As one of the few tropical cities in China, Sanya has gradually developed into one of the most popular coastal tourist resorts. According to the Ma Hive travel data report, Sanya was the most popular tourist destination for the 2022 New Year's Day mini-holiday (*Sheng et al., 2020*).

This article selects Sanya travelogues from the Ctrip and Qunar websites as the data sources for text and images. On the TikTok platform, videos on the topics of ''Sanya'', ''Sanya tourism'', ''Sanya travel'', and ''Sanya travel tips'' are selected as the source of video data.

## Data collection

This article sets the crawler field table to the user ID, travelogue content, publication date, and next-level web link. After analyzing the information required by the crawler, the travelogue is crawled through GooSeeker (http://www.gooseeker.com/). A total of 6,669 original travelogues were eventually crawled, including 1,668 travelogues from Qunar, 5,001 travelogues from Ctrip, and 177,775 original images. The data were collected between November 19 through December 31, 2021, using travel logs.

In this article, for video acquisition, we choose to use the client-side data manager of the advanced version of Collective Searcher, configuring the loading rules, extracting the critical fields (user nickname, publication time, and number of followers and fans), counting the leads and capturing the data. The data are stored as XML files and then converted to Excel format, saving 797 downloadable Jitterbug videos.

## Data preprocessing
### Text splitting and the removal of deactivated words

After the initial screening of travelogues, they are next subjected to word separation. Jieba word separation is commonly used as an open-source word separation tool that

can decompose complete sentences into several independent words, providing a basis for the subsequent feature vectorization of words, latent Dirichlet allocation (LDA) topic model analysis, and statistics of high-frequency words in this article. Deactivated words are mainly noisy information that cannot provide helpful information and seriously affect the accuracy of the analysis results. Deactivated words are mainly personal pronouns, adverbs, or prepositions, such as "is", "many", "directly", and so on. This article collates these deactivation lists, removes duplicate items, and compiles a new deactivation list. Furthermore, expressions and words unrelated to Sanya travelogues are added based on the results of the word separation process.

### Preprocessing of video data

Photos smaller than 30 kb in the image dataset and words and expressions unrelated to the tourist destination are removed, and 138,547 photos are finally saved (*Sheng et al., 2020*). For video selection, users with followers greater than10 are selected (follow rate = the number of followings/the number of followers, where a following rate in the range of 10 to 300,000+ indicates that such accounts are prone to gaining new followers and having a significant impact on their social media sphere (*Keyhole, 2018*). Video processing uses the Python 3.1 interface to call the FFMPEG program. It effectively combines it with the OpenCV module to obtain a video's keyframes, representing the video's main visual content (*Cao, 2013*). FFMPEG is a free, open-source, cross-platform software used mainly to process audio and video files. Most current video-processing research uses OpenCV, an open-source, cross-platform video analysis library used primarily for computer vision and image processing (*Abudukelimu & Wang, 2020*; *Zhang et al., 2021*). The video keyframes are also processed using travelogue images, and a Python deep-learning residual neural network is used to obtain five labels for each keyframe. Keyframes with low confidence and unidentifiable photos are excluded to obtain better confidence levels, and 5,453 keyframe images are finally retained.

## Cognitive image construction
### Constructing a cognitive text image

This article focuses on using LDA to build textual topic models. A topic modeling algorithm is a statistical method that automatically discovers the themes running through unstructured raw text by analyzing its vocabulary. LDA is a commonly used topic modeling algorithm and is a probabilistic generative model for collecting discrete data, such as text corpora. Besides, it is used to discover potential abstract themes in documents or textual data. In the LDA model, the set of documents, number of topics, and hyperparameter after word separation are the model inputs, and the model eventually outputs potential issues. The topic-word distribution under the related topics (where the top-n feature words are arranged from highest to lowest), *etc.* The number of topics extracted by LDA can be calculated using perplexity. Theoretically, the lower the perplexity is, the higher the accuracy of the model prediction would be, and the lowest perplexity or value at the inflection point denotes the appropriate number of topics. Perplexity is an information-theoretic measure, the value of which is defined as energy based on entropy and is often used to compare

probabilistic models. The formula for calculating the degree of confusion is given in Eq. (1).

$$\text{Perlexity}(D) = \exp\left\{-\frac{\sum_{i=1}^{M}\ln p(d_i)}{\sum_{i=1}^{M}N_i}\right\},\qquad(1)$$

### Building visual cognitive images

A deep residual neural network (ResNet), a robust deep learning architecture that won an image classification recognition competition in 2015, is chosen for image recognition. ResNet is a variation of a convolutional neural network that extracts complex features from segmented input data chunks and performs classification through a series of sequential convolutional and fully connected layers. Theoretically, it is generally believed that convolutional neural networks can perform more complex feature pattern extraction than other methods. Moreover, better results are obtained as the number of convolutional layers increases. However, many studies have shown that as the number of layers increases and as the network becomes more profound, the results may worsen because the more profound the network is, the less accurate the classification would be, namely, the performance starts to degrade. ResNet helps resolve the gradient disappearance problem by adding residual blocks to avoid unnecessary convolution and better help the computer with vision tasks to address the abovementioned problem.

To achieve a higher level of confidence in recognizing keyframes extracted from the video, this article uses a k-means algorithm based on the document matrix to reveal the tourist destination image in the tourist footage. The k-means clustering algorithm can determine the clustering results of arbitrary shapes, as it is an unsupervised machine learning method that aims to classify the corpus based on similarities between documents. The document term matrix describes how often terms occur in a collection of documents. The value of this matrix can be expressed as a text frequency-inverse document frequency (TF-IDF) value, which indicates the importance of a term in a document. The k-means clustering algorithm consists of the following steps:

1. select the appropriate number of clusters k;
2. select special k random samples from the input dataset D as the initial prime vector;
3. calculate the distance between the samples and each initial prime in turn, with the samples selecting the closest initial prime as the clustering category;
4. recalculate the prime for each of the calculated clusters;
5. repeat steps 3 and 4 above until k no longer changes in the center of the mass vector;
6. Output the clustering results.

## Sentiment image construction

Sentiment analysis, also known as opinion mining or sentiment propensity analysis, is designed to extract information such as users' emotions and attitudes from the data. This article uses the fine-grained aspect of travelogue text to mine the sentiment intensity of tourists in each travelogue. Extracting tourists' emotional perception attributes concerning Sanya from their travelogues is an essential dimension in the perception of Sanya's tourism

destination image. Two commonly employed sentiment analysis methods are in-text sentiment mining, based on sentiment dictionaries, and machine learning. The machine-learning-based analysis is usually called supervised learning, where a model is trained from a dataset of labeled sentiment scores to obtain the expected results. Eventually, the sentiment probability of each text is calculated. On the other hand, sentiment-lexicon-based analysis is unsupervised learning that determines sentiment tendencies through degree and tone words. As there are no score field data in the original data collection results, this work chooses the SnowNLP method based on sentiment dictionaries. The SnowNLP can easily handle Chinese text content. All algorithms are automatically implemented and can be implemented with some trained dictionaries (*Shang & Zhao, 2021*). This approach's average accuracy, precision, recall, and F value are also better than those of other approaches.

## RESULTS AND DISCUSSION

### Text cognitive image results

According to the results of the LDA output, seen that the word probability distribution of each extracted topic has a long-tail feature. In the tail of the word probability distribution, the word probability values tend to be flat. The subject words with a higher contribution to the top ranking are selected in this article. After the above steps are carried out, the subjects of the travelogue text are extracted, as shown in Table 1. The seven topics are grouped in this article according to the high-frequency words of each issue as follows. The final number of themes identified is 7 ($K = 7$).

The results of the LDA theme model show the textual descriptions of Sanya by tourists on the Ctrip and Qunar platforms. The seven extracted themes can be summarized as "food", "natural environment", "related landmarks", "hotel accommodation and environmental facilities", "recreational activities", "ways to travel and travel prices", and "fellow travelers and ways to travel". Based on the high-frequency words and LDA results, this article summarizes the text's image perceptions into the categories below.

(1) Food perceptions: Hospitality and tourism often complement each other and contribute to each other's development. Shown that food in tourist destinations is gradually becoming an essential component of sustainable tourism. Authentic tourism experiences for tourists explain why local food, especially distinctive food, is a critical attribute that attracts tourists to a destination (*Quan & Wang, 2004*), especially in areas with unique culinary details, such as Sanya. Under the theme of "food", the most frequently mentioned food is 'seafood,' which is in line with the fact that seafood is a local specialty in Sanya, followed by "chicken' and 'coconut" under the food theme. Coconuts and chicken are also key attributes in attracting tourists. More tourists also mention seafood processing than other aspects in their travelogues.

(2) Tourist attractions include mainly natural scenery and related landmarks. Natural attributes and corresponding landmarks are intertwined in the perceived characteristics of tourists. The city of Sanya is blessed with a unique environment and resources, especially the bays of Sanya, such as "Yalong Bay", "Haitang Bay", and "Dadonghai Bay", in addition to "Wuzhizhou Island", "Yanoda Rainforest Scenic Area", and "Hot Springs", which

**Table 1  LDA theme analysis.**

| No. | Topic | Keywords | Proportion |
|---|---|---|---|
| 1 | Foods | Seafood, coconut, restaurant, cuisine, flavour, tofu, taste, fresh, Wenchang chicken, fruit | 10.17% |
| 2 | Natural attributes | Yalong Bay, Wuzhizhou Island, beach, Dadonghai, tropical, Sanya Bay, Haitang Bay, seawater | 48.03% |
| 3 | Related landmarks | Thousands of ancient, scenic, cultural, experience, rainforest, Li, Daxiaodongtian, tropical rainforest, Songcheng, Penang Valley, Yanoda | 10.93% |
| 4 | Hotel accommodation | Hotel, pool, room, restaurant, service, price, environment, free, facilities, attractions, experience | 0.89% |
| 5 | Entertainment | Atlantis, world, aquarium, dolphin, ocean, project, experience, ticket | 1.78% |
| 6 | Excursion price | Price hotel, price, airport, attraction, ticket, public transport, accommodation, free, transportation, driver, cab, guide, duty free | 9.53% |
| 7 | Fellow travellers, Mode of travel, Hotel | Room, beach, breakfast, airport, kids, baby, husband, elderly, parent–child, family, plane, luggage, kids, mom | 18.68% |

together form the perceived attributes of the natural resources of tourism in Sanya. Among the relevant landmarks, tourist trips to attractions with local characteristics and culture, such as "Atlantis", "Li", "Qianguo", the "Penang Valley", "Yanoda", and other tourism resource points, are mentioned and complement the beach resources, along with the promotion of tourism development in Sanya.

(3) Infrastructure: This aspect includes mainly hotel accommodations and environmental facilities. Hotel accommodations are an integral part of tourism activities. A reasonable accommodation environment can provide tourists with quality services and affect their overall perception of the destination. The results of the analysis of thematic high-frequency words show that tourists are concerned mainly with "hotel", "swimming pool", "restaurant", "service", etc. Apart from these factors, tourists' next most important concern is supporting environmental infrastructure.

(4) Tourism leisure and entertainment: These aspects include recreational activities, modes of travel, and prices. In terms of the six elements of tourism, "leisure and entertainment", "Atlantis", and "aquarium" are mentioned more often than the other aspects. Since its opening in 2018, Atlantis has enriched the Sanya water amusement park tourism market, becoming an important attraction for tourists visiting Sanya, with its multi-industry tourism complex integrating a resort, entertainment, catering, conventions, exhibitions, and performing arts.

The last category in the results of extracting themes from travelogue text is that of fellow travelers. The last article shows that tourists travel to Sanya with people with whom they are close, including family and friends. This result aligns with the theme of Sanya being "romantic" and a popular destination for couples. In their travelogues, travelers describe Sanya as a family destination where they travel with their children and elderly family members. Regarding this subject, words related to "children" are frequently mentioned, including children, babies, parent and child relationships, and little children.

## Visual image results of images based on deep learning

In this article, based on the picture classification model of *Zhang, Li & Zhang (2020)*, scene types are divided into six categories—food, accommodation, transportation, travel, shopping, and entertainment—and 11 secondary categories, summarized in Table 2.

According to the classification method of the scenes, this article employs a ResNet to analyze the perception of tourists visiting Sanya. Tourists are most concerned with "tourism", at 46%, followed by "food" and "entertainment", at 23% and 19%, respectively, while housing, transportation, and shopping are of less concern to tourists. Among the 11 secondary subcategories, nature and food are the most attractive to visitors, accounting for 23.80% and 23.18%, respectively, reflecting that Sanya visitors prefer to take photos of nature and food during their travels.

'Tour' accounts for the most significant proportion of the six categories. A related analysis reveals that the number of photos of natural scenery is the highest among tourists' perceptions of 'tour,' accounting for 52.10% of the total photos of 'tour.' This finding is consistent with the brand image of Sanya as a famous tourist city with a tropical seaside. Tourists tend to take pictures of the natural scenery in Sanya. This most important perception is followed by architectural perception, accounting for 34.19% of the total number of "tours", with courtyards and fountains accounting for a relatively important perception. Flora and fauna perception accounts for 10.30% of the total number of "tours", with marine plants accounting for a rather large proportion and cultural perception having the lowest number of photos. Cultural perceptions are the least frequent and weaker than others, accounting for 3.43% of the total "tours". For food perceptions, rice, lobster, crab, and durian are the main attractions, in line with the desire for seafood and tropical fruits among people in Sanya. For 'entertainment,' submarines, sailing, diving, and related clothing are prominent, especially long skirts,' 'swimming costumes,' and 'beach trousers.' In addition, 7.23% of the images are related to wedding themes, which aligns with Sanya's image as a famous wedding photography destination in China. In terms of "travel", the highest number of images are related to "airplane", "port", and "train", indicating that most visitors to Sanya choose to travel by air or boat.

## Results of the visual image of the short video based on keyframe analysis

The k-means clustering of the document matrix is used. The model selection is based on the elbow rule to test different clusters, and the final number of groups is determined to be 6. The arrival of the era of self-media has enriched the ways and platforms through which tourists can share information. Clusters 1 and 2 show that tourists share more information about outbound accommodation attractions, food, and nature in their videos (Table 3). Clusters 3, 4, 5, and 6 are all related to Sanya's natural resources, indicating that Sanya's unique beach and sea areas are considered essential attractions in the tourists' cognitive system. The cognitive attributes associated with these areas are more prominent, which echoes the image of Sanya as a tropical coastal tourist city.

**Table 2** High-frequency vocabulary of tourism image perceptions in Sanya.

| Primary classification | Secondary classification | Frequency | Percentage | Type of scene |
|---|---|---|---|---|
| Eating | Food perception | 32,110 | 23.18% | Restaurant, meal, lobster, vegetable market, crab, fish, durian, corn, banana, ice cream, menu, butcher, cutlery, fish |
| Living | Accommodation perception | 8,071 | 5.83% | Bathtub, bed, sofa, TV, balcony |
| Travelling | Transport perception | 5,976 | 4.31% | Airplane, port, train, tour bus, taxi |
|  | Nature perception | 32,979 | 23.80% | Beaches, sandy beaches, landscapes, coasts, islands, waterfalls, streams, cliffs, volcanoes |
|  | Perception of architecture | 21,651 | 15.63% | Courtyards, buildings, fountains, bridges, sheds, fences, hotels, straw huts, stairs, stone pillars |
| Touring | Flora and fauna perception | 6,523 | 4.71% | Greenery, coral, dogs, birds, sea lions, jellyfish, elephants |
|  | Cultural perception | 2,172 | 1.57% | Letter, bookstore, library |
| Shopping | Shopping perception | 2,344 | 1.69% | Supermarkets, shops |
| Entertainment | Entertainment perception | 27,721 | 19.29% | Submarines, sailing, swings, bands, diving, long dresses, swimming costumes, beach trunks, sunglasses, wedding dresses, hats, weddings, swimming goggles, umbrellas, kimonos, short dresses, flip-flops, binoculars |

**Table 3** K-means clustering analysis.

| No. | Topic | Clustering key tags (in descending order) |
|---|---|---|
| 1 | Scenic architecture | Fountains, planes, landscapes, beaches, swings, bridges, bands, cabinets, bathtubs, buildings |
| 2 | Gastronomy | Dish, restaurant, fish, durian |
| 3 | Nature | Beach, scenery, island, sand, cliff, shore |
| 4 | Beach photography | Beach, swimming costume, long dress, sand, beach trunks, wedding dress |
| 5 | Outfitting | Sunglasses, swimming costumes, swimming glasses, hats, long dresses, umbrellas |
| 6 | Diving & surfing | Diving, coral, swimming glasses, fish, jellyfish, beaches, speedboats |

## The results of the emotional analysis

A good destination image can enhance the tourist experience and lead to positive emotions, while a negative image can lead to a negative experience. The relevant libraries are imported into Python to analyze the texts of Sanya tourists' online travelogues. The results show that tourists' positive sentiment accounts for the highest percentage, 99.39%; negative sentiment accounts for a lower rate, 0.61%. From the analysis, it can be seen that positive emotions dominate the perceived image of visitors to Sanya. Nevertheless, the negative emotions in evaluating some visitors need to be paid great attention.

## Image differences of the tourist destination among multimodal data

First, regarding mental image, tourists' overall perception of Sanya is dominated by natural scenery, human resources, and cuisine, consistent with most studies on Sanya's destination image (*Yin, Yan & Tian, 2019*). The overall emotional image of Sanya's destination is positive. The primary sources of positive emotions include four dimensions: the general

perception of Sanya, natural scenery, holidays, leisure, and culture. Negative perceptions are reflected mainly in value for money (scenic spot tickets, accommodation, etc.) and touring experience (queues, rip-offs, etc.). Visitors' perceptions of Sanya's natural resources, especially coastal resources, cuisine, and tourism and entertainment facilities, and their emotional perceptions constitute their overall perceptions of the destination's image.

Second, specific differences exist between textual and visual images and multimodal data perception images (*Sheng et al., 2020*). The image perception of Sanya tourism destination is grouped into four areas: tourism attraction, leisure and entertainment, infrastructure, and local atmosphere. The textual image is dominated by food, natural attributes, relevant landmarks, hotel accommodations, environmental facilities, recreational activities, outings, tour prices, and travelers and modes of travel, with natural attributes being the central dimension of the textual image and accounting for a more significant proportion.

Scene recognition statistics are identified from six important categories and eleven subcategories in the visual image. In order of preference, the main perceptions are natural scenery, food, entertainment, and architecture. The perception of graphical video images is mainly about outings and entertainment, gastronomy, accommodation, nature, beach photography, dressing, diving, and surfing. The textual and visual perception images are partially similar but with different keywords. Both relate to natural and human attractions, and natural attractions dominate. The visual content appears to be more recreational and leisure activities, such as diving and surfing, than the textual content. The combination of textual and graphic images complement each other and constitute the overall image perceived by tourists. Overall, Sanya's tourism perception image is beautiful natural scenery, unique cuisine, and strong tourism appeal. The overall perceptual image is diverse, positive, and optimistic.

Third, in terms of the overall image, this article summarizes Sanya's tourism destination image into four dimensions: tourism attraction, leisure and entertainment, infrastructure, and local atmosphere based on the hierarchy of sizes by drawing on existing studies (*Xu, La & Ye, 2018*; *Lei et al., 2021*). From the results derived from the text topic probability and scene recognition statistics, it can be seen that there are differences in the dimensions of the image perceptions of the tourism destination. According to the frequency of multimodal data statistics, natural resources are ranked at the first level, tourism, leisure, and entertainment are located at the second, infrastructure is put at the third, and the local atmosphere is ranked at the fourth.

## CONCLUSION

This article makes several significant theoretical contributions to the literature. First, this work extends the line of research on destination images by exploring the destination images implied in various data under UGC multimodal big data and using research methods such as deep learning and machine learning. Visual information can attract tourists' attention (*Somabhai, Varma & Somabhai, 2015*), generate perceived destination images (*Wang & Sparks, 2016*; *Deng et al., 2019*; *Adel & Hamed, 2019*), and influence travel decisions (*Xiao et al., 2022*). However, visual information is not well recognized in studying

the tourism destination image. Most extant literature still analyzes tourism destination images from unimodal data (*Tan, Liu & Li, 2021*; *Zhang, Chen & Li, 2019*). Therefore, the results of this article significantly enrich the tourism literature and provide new findings for the study of the visual content of tourists. For example, this article shows differences between textual and visual cognitive images regarding perceptions of tourism destination images. In addition, information such as tourism photographs and videos contains different perceptual content that distinguishes it from text. Second, this article adds the range of short videos by Shake Shack to the study of tourism visuals to explore and analyze the elements of tourism destination images contained in tourist-generated videos. At the same time, the research on tourism destination image in the era of big data has been continuously deepened by accurately identifying the massive tourism visual content through artificial intelligence (*Yu & Egger, 2021*; *Kuhzady & Ghasemi, 2019*; *Vu et al., 2015*).

On the other hand, the established studies on visual tourism content focus mainly on picture information (*Kuhzady & Ghasemi, 2019*; *Hu et al., 2015*; *Felbermayr & Nanopoulos, 2016*), ignoring the vital information of short videos. In this article, the Python interface was used to call the FFMPEG program and effectively combined it with the OpenCV module to obtain the keyframes of the video to explore the main visual content. Techniques like k-means clustering based on the document term matrix and ResNet are mixed. As this article analyses review images and shake videos in Ctrip and other platforms, the proposed model is already a generalized algorithm that can be applied to any data in the form of images. This study explores the possibility of employing machine learning tools to discover potential insights in the data and provides new directions for future video data processing in the tourism domain.

### Funding
This research was funded by the the National Natural Science Foundation of China (under no. 72162014). The funders had no role in study design, data collection and analysis, decision to publish, or preparation of the manuscript.

### Grant Disclosures
The following grant information was disclosed by the authors:
National Natural Science Foundation of China: 72162014.

### Competing Interests
The authors declare there are no competing interests.

### Author Contributions
- Tao Hu conceived and designed the experiments, performed the experiments, analyzed the data, performed the computation work, prepared figures and/or tables, authored or reviewed drafts of the article, and approved the final draft.

- Juan Geng conceived and designed the experiments, performed the experiments, analyzed the data, performed the computation work, prepared figures and/or tables, authored or reviewed drafts of the article, and approved the final draft.

## Data Availability

The code is available in the Supplemental File.

The data is available at Zenodo: DGU-AI-LAB. (2019). DGU-AI-LAB/Korean-Tourist-Spot-Dataset: Korean Tourist Spot Dataset (v0.1) [Data set]. Zenodo. https://doi.org/10.5281/zenodo.3381859.

## Supplemental Information

Supplemental information for this article can be found online at http://dx.doi.org/10.7717/peerj-cs.1801#supplemental-information.

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
