# Peer review of "Research on the perception of the terrain image of the tourism destination based on multimodal user-generated content data"

_PeerJ Computer Science, doi:10.7717/peerj-cs.1801_

## Round 0.1 · original submission · Major Revisions

Dear Authors,

Your submission has been reviewed by experts in the field and you will see that they are suggesting a lot of changes for the improvement of your article. So, please make sure the response to all the comments in your revised paper so that we can consider it further.

Moreover, Please also justify the contribution of your work and revise the abstract to include more information about methodology and results.

Finally, Also improve the technical language of your article.

**Language Note:** The Academic Editor has identified that the English language must be improved. PeerJ can provide language editing services - please contact us at [email protected] for pricing (be sure to provide your manuscript number and title). Alternatively, you should make your own arrangements to improve the language quality and provide details in your response letter. – PeerJ Staff

Reviewer 1 ·

Basic reporting

Even though the current article has contributed to the available literature, there exist some severe issues that need to be responded to by the authors completely. The detected issues are itemized as follows:
1. The title of the article contains an abbreviated word. It should be provided fully.
2. the introduction is short. It should be expanded with more discussion and references.
3. the research motivation and the contribution should be expressed in a separate paragraph in the introduction section
4. the outline of the paper should be stated in the introduction section
5. The abstract should be rewritten and reorganized with more focus on the key findings.
6. All citations should follow a regular pattern.
7. Abbreviated words should be used in the text, not in the titles.
8. All abbreviated words should be checked and fixed.
9. the titles of all sections and subsections should be checked.
10. The version of the software should be given
11. All equations should be cited in the text.
12. Some paragraphs are very long. They need to be shortened.
13. The conclusion section should be constructed.
14. Why K-means clustering is run since a better clustering algorithm should be used instead? Please discuss it.
15. How is the number of clusters determined by the authors? Please discuss it.
Thus, I recommend a major revision.

Experimental design

.

Validity of the findings

.

Additional comments

I wish to see the revised version again.

Reviewer 2 ·

Basic reporting

User-generated data is used to derive some insights into a destination called Sanya in China. It is important research with substantial outcomes and practical implications. However, the authors did not present some key arguments and some technical aspects are not clear to the reader. I underlined them as follows:
1. The author should more clearly and precisely define the problem statement in the abstract of the paper “what scientific problem authors addressed in this paper?”.
2. Just like above comments, I would suggest authors to summarize the contributions of this paper at the end of Introduction Section. For example, author may write: “This paper has the following contributions:”
3. Section 3 is “Research Framework Design”, and authors showed their design in Figure 1. I would suggest authors to update the Figure 1 with some hints on all arrows as well lines that are linking the different blocks in the figure.
4. Even though, in section 4, authors have presented the results and analysis but I would suggest that mathematical analysis would give more confidence to the readers.
5. Did the authors eliminate some data? What was its proportion? Please discuss it.
6. Did the authors train the model and test it? Please discuss it. What are the ratios of the test and training data?
7. Can the derived results be expanded to similar resorts in China? Please discuss it.
8. K-mean clustering is though widely implemented method, but it has several faults. How did they deal with it? Why did the authors use a more robust one?
9. How did the authors decide to have 6 clusters? Please discuss.
10. What software and package are used to derive results? Please discuss.
11. The authors should construct those sections: 1. The conclusion, 2. Practical implications, 3. Theoretical implications.
12. The authors use a data set containing various data sources (multimodality). Did they reach a result different than the results derived from a relatively limited data set? Please discuss.
13. Both the results section and conclusion section should be constructed.
14. Even though the level of the English language is acceptable, improvement is needed.
15. Did the authors detect by chance a conflict between modalities of the data? Please discuss.
16. Did the authors find similar patterns or different ones when the results were compared with the similarly conducted research in the literature? Please discuss it.
17. Instead of presenting generic remarks, more research-based sentences should be used to enrich the article.

Experimental design

.

Validity of the findings

.

---

## Round 0.2 · accepted · Accept

Thanks for your revised submission in light of the comments of the experts,
I'm happy to inform you that experts has now commented and they are happy to accept your article. I endorse their decision. Congratulations and thank you for your contribution

Reviewer 1 ·

Basic reporting

The authors carefully incorporate the suggested changes in the revised version of the manuscript. So I agreed with the manuscript in its current form for publication.

Experimental design

The authors carefully incorporate the suggested changes in the revised version of the manuscript. So I agreed with the manuscript in its current form for publication.

Validity of the findings

The authors carefully incorporate the suggested changes in the revised version of the manuscript. So I agreed with the manuscript in its current form for publication.

Reviewer 2 ·

Basic reporting

I think the author has answered most of the comments given in the first review.

Experimental design

I think the author has answered most of the comments given in the first review.

Validity of the findings

I think the author has answered most of the comments given in the first review.

Additional comments

I think the author has answered most of the comments given in the first review.